# A genomic basis of vocal rhythm in birds

Matteo Sebastianelli [1,2] ✉, Sifiso M. Lukhele [1], Simona Secomandi [1], Stacey G. de Souza[1], Bettina Haase[3], Michaella Moysi[1], Christos Nikiforou[1], Alexander Hutfluss[4], Jacquelyn Mountcastle[3], Jennifer Balacco[3], Sarah Pelan[5], William Chow[5], Olivier Fedrigo [3], Colleen T. Downs [6], Ara Monadjem [7,8], Niels J. Dingemanse [4], Erich D. Jarvis [3,9,10], Alan Brelsford[11], Bridgett M. vonHoldt [12] & Alexander N. G. Kirschel [1] ✉

Vocal rhythm plays a fundamental role in sexual selection and species recognition in birds, but little is known of its genetic basis due to the confounding effect of vocal learning in model systems. Uncovering its genetic basis could facilitate identifying genes potentially important in speciation. Here we investigate the genomic underpinnings of rhythm in vocal non-learning *Pogoniulus* tinkerbirds using 135 individual whole genomes distributed across a southern African hybrid zone. We find rhythm speed is associated with two genes that are also known to affect human speech, Neurexin-1 and Coenzyme Q8A. Models leveraging ancestry reveal these candidate loci also impact rhythmic stability, a trait linked with motor performance which is an indicator of quality. Character displacement in rhythmic stability suggests possible reinforcement against hybridization, supported by evidence of asymmetric assortative mating in the species producing faster, more stable rhythms. Because rhythm is omnipresent in animal communication, candidate genes identified here may shape vocal rhythm across birds and other vertebrates.

Research on the mechanisms that shape and maintain biological diversity is one of the hot topics in evolutionary biology, with non-model organisms progressively improving our understanding of the molecular basis of phenotypic traits[1]. Among other systems[2,3], research on birds has provided crucial insights into the mechanisms underlying phenotypic variation as well as the molecular basis of adaptation[4]. In particular, feather color has attracted much attention because of its importance in mate choice and speciation, with genes identified that function in melanin pigmentation[5] and in carotenoid conversion from yellow to red feather color among species[6] and between the sexes[7]. Recent work has even found genes associated with differences in distance traveled by migratory birds[8,9]. Notwithstanding, identifying

genes that function in behavior is challenging, partly because of the role of cultural learning in animal behavior.

Bird song is a highly labile trait, affected greatly by anatomy, the environment[10], and in most species by cultural learning[11,12]. Indeed, most genetic studies on bird song have so far focused on the genes implicated in song learning[13–16], thus leaving the molecular mechanisms underlying the temporal or spectral patterning of bird songs unexplored. In particular, rhythm, which entails song elements regularly distributed in time[17,18], constitutes an essential component of animal communication that mediates key social behaviors[18,19], such as individual recognition[20] and mate selection[21–23]. Although comparative approaches have contributed towards a better understanding of the

[1]Department of Biological Sciences, University of Cyprus, PO Box 20537 Nicosia 1678, Cyprus. [2]Department of Medical Biochemistry and Microbiology, Uppsala University, Box 582, 751 23 Uppsala, Sweden. [3]Vertebrate Genome Lab, The Rockefeller University, New York, NY, USA. [4]Behavioural Ecology, Faculty of Biology, LMU Munich (LMU), 82152 Planegg-Martinsried, Germany. [5]Wellcome Sanger Institute, Cambridge, UK. [6]Centre for Functional Biodiversity, School of Life Sciences, University of KwaZulu-Natal, Pietermaritzburg 3209, South Africa. [7]Department of Biological Sciences, University of Eswatini, Kwaluseni, Eswatini. [8]Mammal Research Institute, Department of Zoology & Entomology, University of Pretoria, Private Bag 20, Hatfield, 0028 Pretoria, South Africa. [9]Laboratory of Neurogenetics of Language, The Rockefeller University, New York, NY, USA. [10]Howard Hughes Medical Institute, Chevy Chase, MD, USA. [11]Department of Evolution, Ecology and Organismal Biology, University of California Riverside, Riverside, CA 92521, USA. [12]Department of Ecology & Evolutionary Biology, Princeton University, Princeton, NJ 08544, USA. ✉e-mail: matteo.sebastianelli@imbim.uu.se; kirschel@ucy.ac.cy

functions of rhythm processing[17,24], it is less clear how rhythmic differences arise and how these are maintained over time. Bird song frequency has a tight relationship with body mass[25,26], and is likely polygenic[27]. Temporal features of song, such as pulse rate, by contrast, may involve signal transduction, neural regulation, or physiological constraints of the syrinx or lungs[28]. Although rhythmic components have been described in birds[24], humans[29], and non-human primates[30], it is only in humans that a molecular basis of rhythmic ability has been identified[29].

Here, we investigated the genetic basis of bird song rhythm in two vocal non-learning species of *Pogoniulus* tinkerbirds, African barbets (Piciformes: Lybiidae), that are widely distributed across Sub-Saharan Africa (Fig. 1a) and emit remarkably simple rhythmic songs[26]. Their songs comprise a repetitive series of pulses delivered at constant pitch and rate, with the latter differing subtly but unambiguously between Southern African populations of two species, yellow-fronted tinkerbird (*P. chrysoconus extoni*, hereafter *extoni*) and red-fronted tinkerbird (*P. pusillus pusillus*, hereafter *pusillus*) (Fig. 1b, Supplementary Fig. 1, and Supplementary Movies 1–3). Intriguingly, although *extoni* and *pusillus* hybridize extensively when they come into secondary contact[31] (Fig. 1c, d), evidence suggests introgression is asymmetric[32]. Due to the simple structure of the vocal phenotype that is divergent between the two species and the extent of introgression between them[32], this

system provides the opportunity for a genome-wide association study (GWAS) to identify the genetic basis of the rhythmic components of tinkerbird song, and we also consider the extent of dominant or additive inheritances of candidate regions.

We quantified rhythmicity in tinkerbird song by investigating the presence of categorical rhythms[24], in which time intervals between note onsets are distributed categorically rather than continuously[33,34]. Using a reference genome for *P. pusillus* we assembled, we then investigated which areas of the genome are associated with variation in vocal rhythm. We focused specifically on the inter-onset interval (IOI): i.e., the time-interval between the onset of consecutive pulses, and thus a measure of pulse rate. It is widely recognized that spectral[10,26], but also temporal, characters of bird song[35] may be shaped by the environment. We thus also tested for the effect of the environment on IOI. Moreover, we needed to establish whether both sexes sing. Tinkerbirds are sexually monomorphic, and unlike several other species in Lybiidae, do not duet[36], and any possible variation in IOI attributable to sex would need to be accounted for.

Having identified the genomic regions associated with IOI, we explored whether they are under directional selection using long-range haplotype statistics. We then revealed their association with genome-wide and candidate gene-specific local ancestry to determine their relative effects on IOI and stability by hypothesizing that (1)

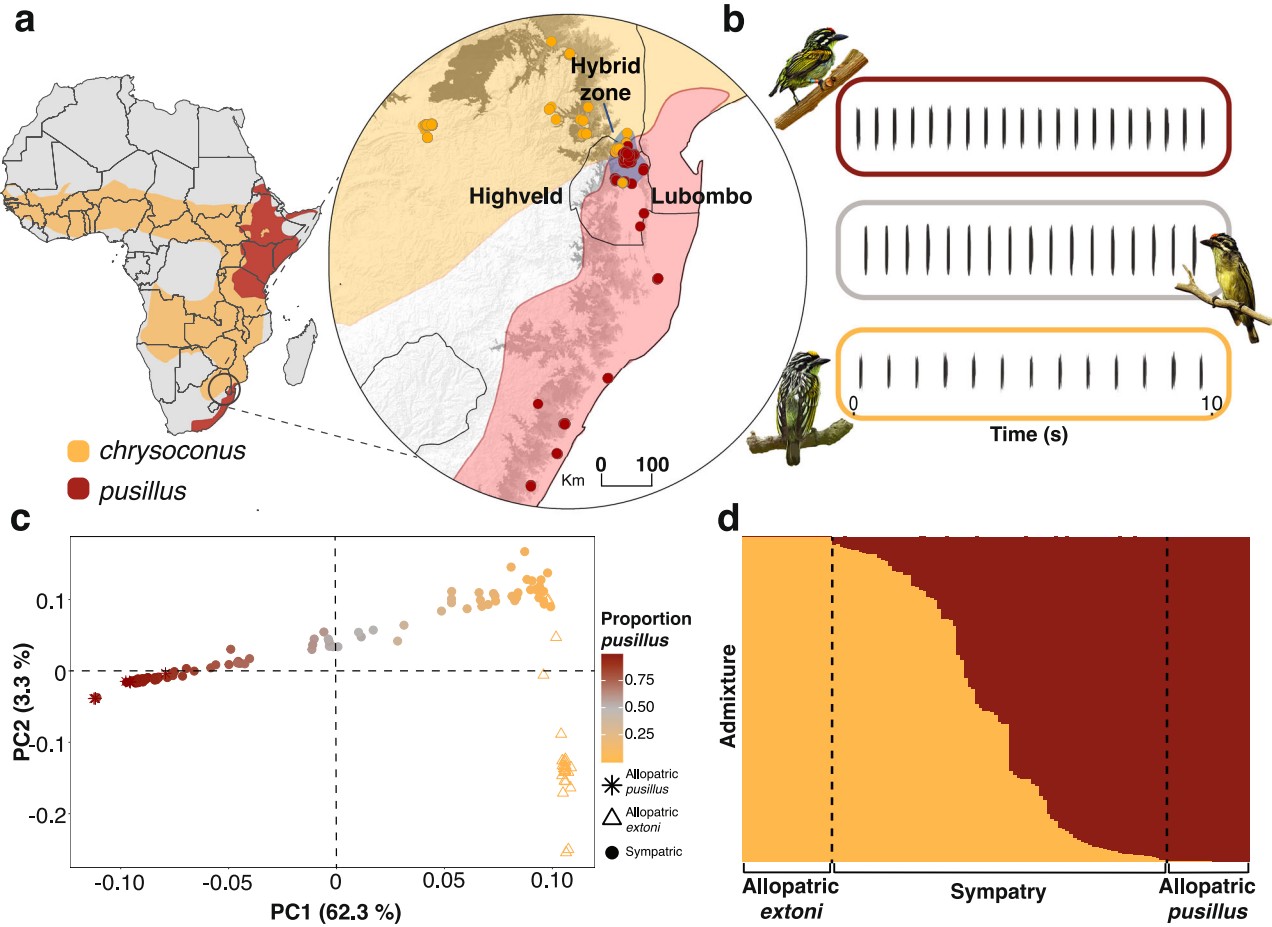

**Fig. 1 | Distribution, phenotype, and hybridization patterns of yellow-fronted (*extoni*) and red-fronted tinkerbird (*pusillus*). a** Geographic distribution of *P. chrysoconus* (yellow) and *P. pusillus* (red) across Africa (basemap available at: https://databasin.org/datasets/23ac736398d84d09aecf3e6760954314/), with insert focusing on the narrow hybrid zone (blue shading), channeled between the western highlands (highveld) and Lubombo mountains. Circles represent sampling localities of 452 tinkerbirds, with colors representing species according to forecrown color. **b** Schematic spectrograms of tinkerbird song (see Supplementary Fig. 1a for full spectrograms), illustrating rhythmic rate differences between individuals of the two species and intermediate song in a hybrid (gray border). **c** Whole-genome PCA, color-coded by ancestry with gray representing admixed individuals, and **d** ADMIXTURE plot (K = 2), together reveal the extent of mixed ancestry in the contact zone.

hybrids may sing intermediate or more variable pace song than pure ancestry individuals as a consequence of their admixed genomes and that (2) such hybrid phenotypes might be considered unattractive by opposite-sex individuals in the contact zone. We considered these hypotheses in the context of possible asymmetric hybridization in the contact zone. Besides pulse rate, other traits may lead to a nonrandom mating pattern, such as body size[37], which in turn correlates with song frequency[25], and feather color. Hence, we investigated the covariance of IOI with these other phenotypic traits in our study system.

We identified a cluster of genes on one chromosome associated with IOI, including two genes previously associated with vocal dysfunctions and one with hearing disabilities in humans. We further found that locally inferred genetic ancestry at these loci was associated with IOI and its stability, and that there is a striking pattern of asymmetric nonrandom mating in the contact zone, which is consistent with a female *pusillus* preference for males with higher proportions of *pusillus* ancestry; i.e., those males that sing faster and more stable songs. Our findings provide insights into the genetic architecture of bird song and on the processes that contribute to phenotypic variation of potential importance in reproductive isolation.

## Results

### Equally spaced IOIs reveal isochrony in tinkerbird song

Using 80,019 IOI measurements taken from recordings of 124 allopatric male individuals, we calculated the rhythmic ratios ($R_k = IOI_k/(IOI_K + IOI_{K+1})$) for *extoni* ($n = 81$) and *pusillus* ($n = 43$). Although mean IOI differed significantly between the two species (*extoni* = -0.57 s ($\pm 0.06$ SD), *pusillus* = 0.46 s ($\pm 0.02$ SD) t-test: $t = 338.04$, $df = 65517$, $P = <0.001$), their $R_k$ did not differ (t-test: $t = 0.72$, $df = 22166$, $P = 0.468$), with *extoni* $R_k = 0.50$ ($\pm 0.01$ SD) and *pusillus* $R_k = 0.50$ ($\pm 0.008$ SD). Tinkerbird songs are, therefore, isochronous, with IOI reflecting the on-integer 1:1 ratio (Supplementary Fig. 2), similar to a metronome's tempo[38], demonstrating that pulses are delivered at intervals of identical duration, a feature that has thus far been uncovered only in birds with vocal learning[24,39]. We then investigated the genetic basis of this rhythmic feature (i.e., IOI).

### A newly assembled reference genome for *Pogoniulus pusillus*

We assembled a chromosome-level reference genome for a female *pusillus* (GenBank accession number: GCA_015220805.1) using the Vertebrate Genomes Project (VGP) pipeline v1.6. This included a combination of PacBio CLR long reads, 10x Genomics linked-reads, Bionano optical maps, and Hi-C data[40] (see Methods). The final assembly was 1.27 Gb in length (Supplementary Fig. 3a and Supplementary Table 1). We produced an assembly with a scaffold NG50 of 26 Mb and N50 of 34 Mb (Supplementary Table 2), a contig N50 of 16.8 Mb, and a per-base consensus accuracy (QV)[41] of 42.8 (-0.53 base errors/10 Kbp; Supplementary Table 3). The assembly has a GC content of 46.0% (Fig. 2), a repeat content of 47.3%, a functional completeness[42] of 95.2% (Supplementary Table 4), and a k-mer completeness[41] of 85.2% (93.9% when combined with the alternate assembly; Supplementary Fig. 3b and Supplementary Table 3). We assigned 97.8% of assembled sequences to 44 autosomes and the sex chromosomes, Z and W ($2n = 90$; Fig. 2 and Supplementary Fig. 3c). The karyotype is concordant with other Piciformes[43].

### The genetic architecture of rhythm in tinkerbirds

We collected whole-genome sequence data with an average 8.2-fold coverage from 138 color-banded tinkerbirds that were aligned to our newly assembled reference genome using BWA-MEM v5.6.1[44] and 19.6 million single nucleotide polymorphisms (SNPs) were discovered using GATK4[45]. We conducted a GWAS using a linear mixed model (LMM) implemented in GEMMA[46] to identify nonrandom associations between IOI and SNPs generated for 87 individuals in the hybrid zone. Focusing only on genomic regions with multiple significant SNPs, we

discovered a region strongly associated with IOI variation (Fig. 3a). This region spans -13 Mb of chromosome 25 (Fig. 3b, c) and is defined by 15 significantly associated SNPs (−log10P > 6) including three that passed a more stringent threshold of −log10P > 7 (Supplementary Table 5). We annotated these outlier SNPs by aligning the tinkerbird reference to the annotated zebra finch (*Taeniopygia guttata*) reference (RefSeq assembly accession: GCF_003957565.2) with BLAST v2.9[47]. All 15 significant SNPs mapped onto zebra finch chromosome 3, with three SNPs annotated in introns of Neurexin 1 (*NRXN1*), one SNP in an exon of Coenzyme Q8A (*COQ8A*) although not in a coding sequence, one SNP in an intron of ENAH actin regulator (*ENAH*) -100 bp from a coding sequence in intron 2, and one SNP in an intron of MutS homolog 2 (*MSH2*) (Fig. 3b and Supplementary Table 6). Given the pattern of variance partitioning, there is likely an uneven contribution from each of the four candidate genes (Fig. 3d), with three genes (*NRXN1, COQ8A,* and *MSH2*) each providing significantly greater contributions to total trait variance than *ENAH* and 10,000 randomly selected SNPs across the genome (Supplementary Table 7). Besides these four primary candidate genes, we highlight five additional SNPs that may be involved in the transcriptional regulation of the following genes, given their proximity to them (within 10 Kb 3' or 5' from the closest gene[48]): a single SNP located 69 bp upstream of an undescribed gene whose zebra finch orthologs are yet to be identified (LOC115494518), one SNP within 1 Kb downstream of anaplastic lymphoma kinase (*ALK*), two SNPs located -9.4 and 15.3 Kb upstream of Forkhead box N2 (*FOXN2*), and one SNP located 9.8 Kb upstream of echinoderm microtubule-associated protein like-4 (*EML4*).

An in-depth analysis of the genetic architecture of IOI using a Bayesian sparse linear mixed model (BSLMM) indicated that 95.6% of the phenotypic variance was explained by the complete set of -19 M analysed SNPs (PVE, or proportion of variance explained by both sparse and random effects), of which sparse effect terms (i.e., PGE: alleles with large phenotypic effect) accounted for 47.3%. We further extracted 76 SNPs that have the largest sparse effect on IOI (i.e., sparse effect >99.9% quantile). Of these 76 SNPs, 12 were located on chromosome 25 and explained 15.78% of the total effect size (based on those 67 SNPs of large effect). Interestingly, the relative contribution of the SNPs on chromosome 25 increases when using a more stringent threshold: when extracting SNPs with a sparse effect above 99.99% quantile, SNPs on chromosome 25 accounted for -49.25% of the total effect size. Whatever the approach, the SNP with the largest sparse effect is rs9449012, which maps onto an intron of *MSH2* (Table S5, S6). This SNP is also the one that has the strongest association with IOI (highest posterior inclusion probability, or the frequency to have a detectably large effect). We further evaluated the ability of our SNP dataset to accurately predict the vocal phenotype with a Bayesian sparse linear mixed model. Using leave-one-out cross-validation, we found that the predicted and observed IOI trait values were highly correlated (Pearson's $R = 0.78$) and that the inferred phenotypes explained a statistically significant proportion of observed IOIs (LM: $\beta = 2.04$, st. error = 0.17, $t = 11.53$, $P = <0.001$, adjusted $R^2 = 0.6$; Supplementary Fig. 4).

We also calculated long-range haplotype statistics to identify potential signatures of directional selection. We estimated cross-population extended haplotype homozygosity[49] on allopatric individuals of either parent species but found that no significant SNP deviated beyond the 2.5 and 97.5% quantiles of the score distribution. However, most SNPs carried a negative value, possibly revealing weak or ongoing selection in *pusillus* (Fig. 3e).

### Population structure and the genomic landscape of differentiation and recombination

Within the -13 Mb region spanned by our significant SNPs, we identified a region of -2.7 Mb with relatively high $F_{ST}$ levels compared to the background of chromosome 25 ($F_{ST}$ within peak = $0.23 \pm 0.06$, $F_{ST}$

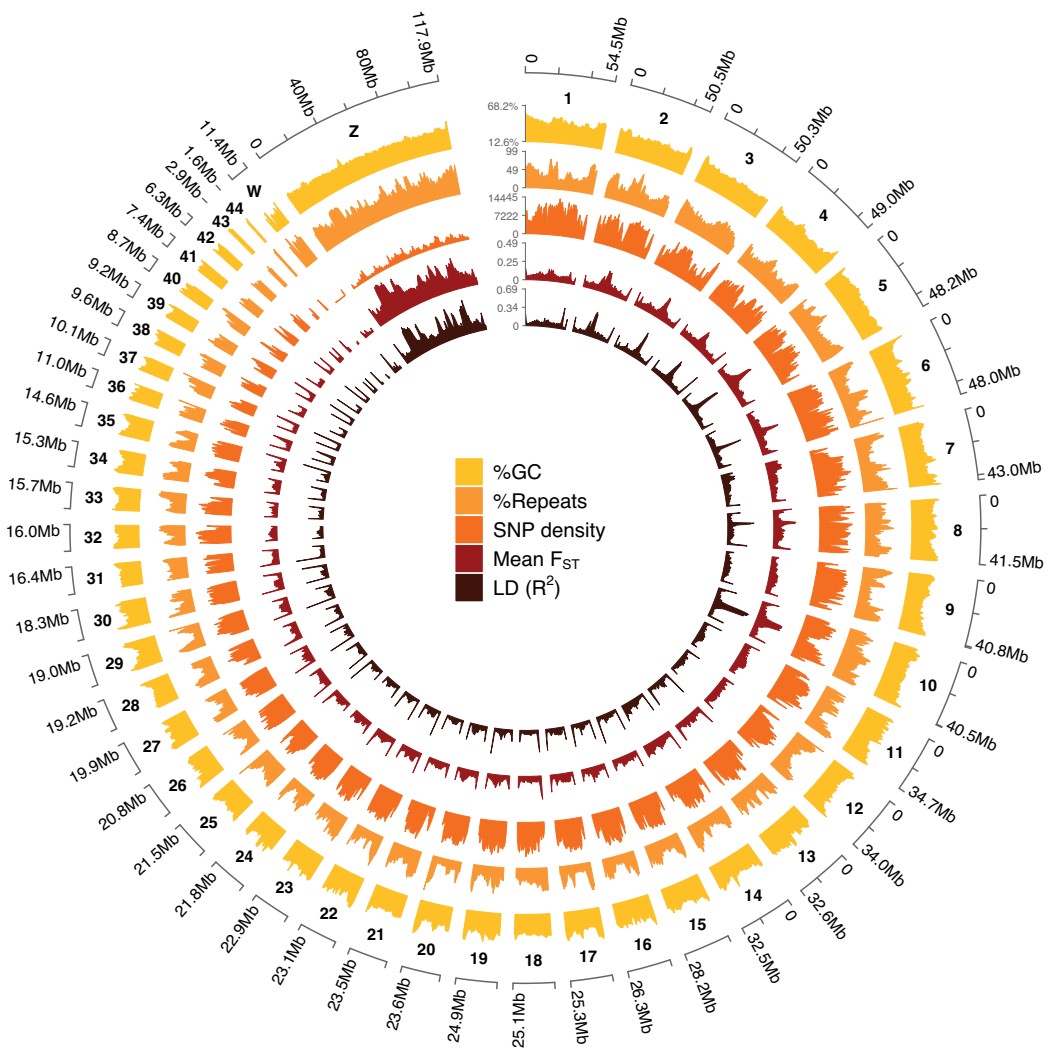

**Fig. 2 | The *pusillus* reference genome.** Circos plot representing the reference genome chromosomes. Data were plotted using 500 kbp windows. For each window, the percentage of G and C bases (%GC), the percentage of bases masked with Windowmasker and Repeatmasker (%Repeats), the number of SNPs (SNP density), the mean $F_{ST}$ value (mean $F_{ST}$), and the mean LD value (LD ($R^2$)) is reported.

outside peak = 0.18 ± 0.05, *t*-test: $t = -8.658$, $df = 134.58$, $P = <0.001$) between allopatric *extoni* and *pusillus* (Fig. 3f), even though $F_{ST}$ within the peak was not different from genome-wide levels (*t*-test: $t = 6.794$, $df = 110.22$, $P = <0.001$). We then compared our findings for $F_{ST}$, a measure of relative differentiation, with those for $D_{XY}$, a measure of absolute differentiation, and investigated potential causes for the observed landscape of differentiation. A striking pattern of discordance within the divergence peak emerged from this comparison around 10 Mb, in proximity to *MSH2*. Such discordance may be the consequence of a reduction in genetic diversity (Fig. 3h), which can be caused either by recent ecological selection or by ongoing background selection[50] and tends to occur in areas of relatively low recombination rates, as shown in our results (see Fig. 3h and Supplementary Fig. 5a). Beyond the major island of divergence, higher relative $F_{ST}$ coupled with higher $D_{XY}$ occurred in two regions flanking the main divergence peak: one between 6 and 8 Mb, where *NRXN1* and *COQ8A* SNPs are located, and one between ~11.5 and ~14.5 Mb. Such covariance between $F_{ST}$ and $D_{XY}$ in areas of relatively high differentiation is expected to be caused either by recent gene flow or under ancient divergence of haplotypes[50]. Following ref. 51, we attempted to disentangle the two scenarios by running $F_{ST}$ and $D_{XY}$ scans in the sympatric population by focusing on *extoni* and *pusillus* individuals with pure ancestry >95% (Supplementary Fig. 5b, c). We assumed that if gene flow was

responsible for the formation of the genomic islands, then scans performed in the sympatric population would show similar levels of $F_{ST}$ and $D_{XY}$ in such regions and reduced $D_{XY}$ outside these islands as a consequence of gene flow. However, we did not observe more pronounced $D_{XY}$ in the scans performed with the sympatric individuals, a result that is consistent with ancient divergence of haplotypes[50,51].

We also observed higher nucleotide diversity in *extoni* (Supplementary Fig. 6 and Supplementary Table 9) as well as little evidence of linkage disequilibrium, with mean $r^2 = 0.09$ on chromosome 25 (mean genome-wide $r^2 = 0.11$) (Supplementary Fig. 7). This suggests that the SNPs in the four candidate genes associated with IOI may not have been inherited together as a linkage group. This might not be the case though, for genes found within the ~2.7 Mb locus containing *MSH2* (see Table S5), which is characterized by a drop in recombination rates (Supplementary Fig. 5a) below genome-wide levels.

In addition, of the genotyped SNPs that mapped directly onto described genes, four were fixed in allopatric populations of *pusillus* and three in *extoni* (three on *NRXN1* and one on *COQ8A*) but all were variable in sympatry, with IOI reflecting underlying genotypes at those loci (Fig. 3i and Supplementary Fig. 8), consistent with additive effects rather than dominant inheritance. Although there was a detectable population structure between *extoni* and *pusillus* when found in allopatry, the extent of genomic admixture in the contact zone was

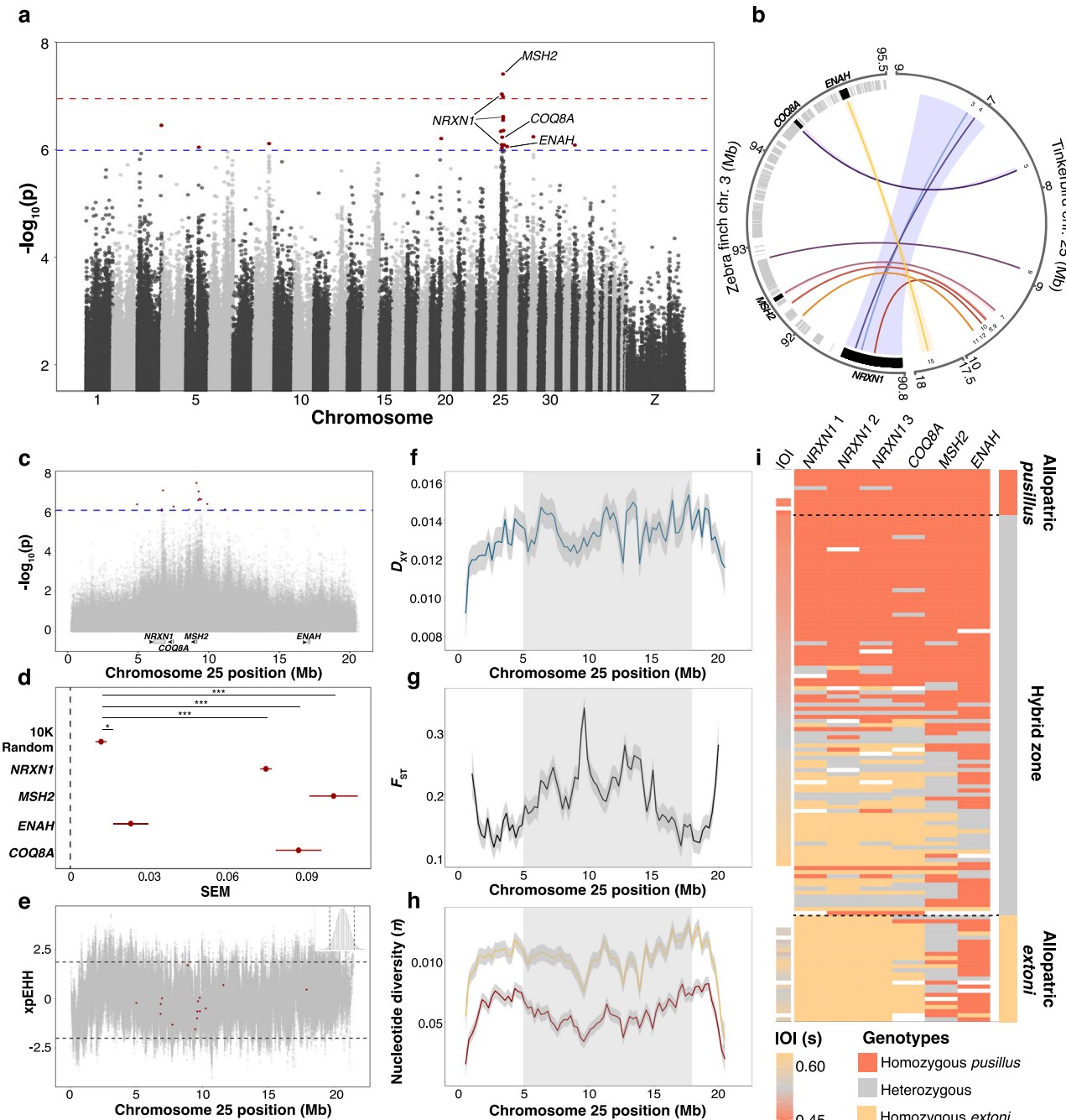

**Fig. 3 | Candidate genes and genomic scans for signatures of selection and diversity. a** Manhattan plot showing regions of the genome significantly associated with IOI, with red dots illustrating significant SNPs resulting from LMM analysis in GEMMA and dashed lines the significance thresholds set by permutation. **b** Illustration of the exact location where the 15 significant SNPs map on chromosome 3 of the zebra finch (thin link lines). The zebra finch genes falling in the region are represented with gray squares. The four candidate genes are in black. Shaded links represent the correspondence between the whole zebra finch gene and *pusillus* chr. 25. **c** Close-up of (**a**) representing the location of significant SNPs (red dots, from LMMs in GEMMA) and candidate genes on chromosome 25. **d** The relative contribution of significant SNPs to the variance explained (mean and SEM based on 15,940 SNPs in *NRXN1*, 995 SNPs in *COQ8A*, 1317 SNPs in *MSH2*, 1210 SNPs in *ENAH* and 10,000 randomly selected SNPs). Asterisks refer to significant *p* values as resumed in Supplementary Table 7, with '*' indicating $p > 0.001$ and '***' $p < 0.001$. **e** Main output of xpEHH on allopatric individuals, with 15 SNPs associated with IOI illustrated in red, and positive (selection for *extoni*) and negative (selection for *pusillus*) significance thresholds. Means and 95% CI illustrate variation in (**f**) $D_{XY}$ and (**g**) $F_{ST}$, and (**h**) comparison of π between the two species across chr. 25 (the gray shaded area spans the range of the significant SNPs). **i** Candidate gene SNP genotypes across 138 individuals and associated IOI for those individuals whose songs were recorded. Note that diversity statistics in (**e**–**h**) refer to allopatric individuals only.

evident in both the principal component analysis (PCA) and ADMIX-TURE (Fig. 1c, d). Also, a PCA of the individual SNPs associated with each candidate gene separately revealed three discrete clusters, one of which carried mostly the heterozygous genotype at candidate loci (Supplementary Fig. 9).

## Pulse rate is not affected by habitat or sex

In addition to genetic effects, the environment may also shape pulse rate[35]. We tested for the effect on IOI of vegetation density, measured using the Enhanced Vegetation Index (EVI) extracted from our recording localities in the contact zone. We found no effect of EVI on

IOI after controlling for autosomal ancestry and location (Supplementary Table 8). Furthermore, using double-hierarchical generalized linear models (DHGLMs), we found no IOI differences between the sexes. Having sexed color-banded birds we recorded (see Relative sequence depth sexing), we established that our dataset included seven recorded females from the contact zone (Supplementary Table 15), but neither IOI, nor its residual within-individual variance (RWV, i.e., level of stability in IOI - see description in next section) differed between males and females.

## Local ancestry is associated with IOI and its variability

We inferred genome-wide global and local ancestry of 99 individuals sampled from within the geographic hybrid zone, which ranged from one parental type (*extoni* $Q_{pusillus}$ = 0.03) to the other (*pusillus* $Q_{pusillus}$ = 0.99) (Fig. 4a). Across the hybrid population, we estimated a

mean of $60 \pm 36\%$ *pusillus* ancestry with genomes structured as a mosaic of ancestry blocks (Fig. 4b). Introgression of ancestry blocks between the two lineages at this contact zone has been ongoing for hundreds of years (autosomes = 274.11 years, Z chromosome (Z/W sex determination system) = 429.3 years for the oldest admixture event; autosomes = 37.4 years, Z chromosome = 75 years for the most recent introgression between non-admixed individuals of the two species), with hybrid individuals containing smaller sized ancestry blocks the older the admixture event (Fig. 4c), consistent with an effect of recombination in breaking down ancestry blocks across generations. As a consequence, admixed individuals have many more and smaller ancestry blocks than pure ancestry individuals (Supplementary Fig. 10a, b).

Further corroborating the association we report between IOI and specific SNPs, we find a relationship between ancestry and IOI revealed

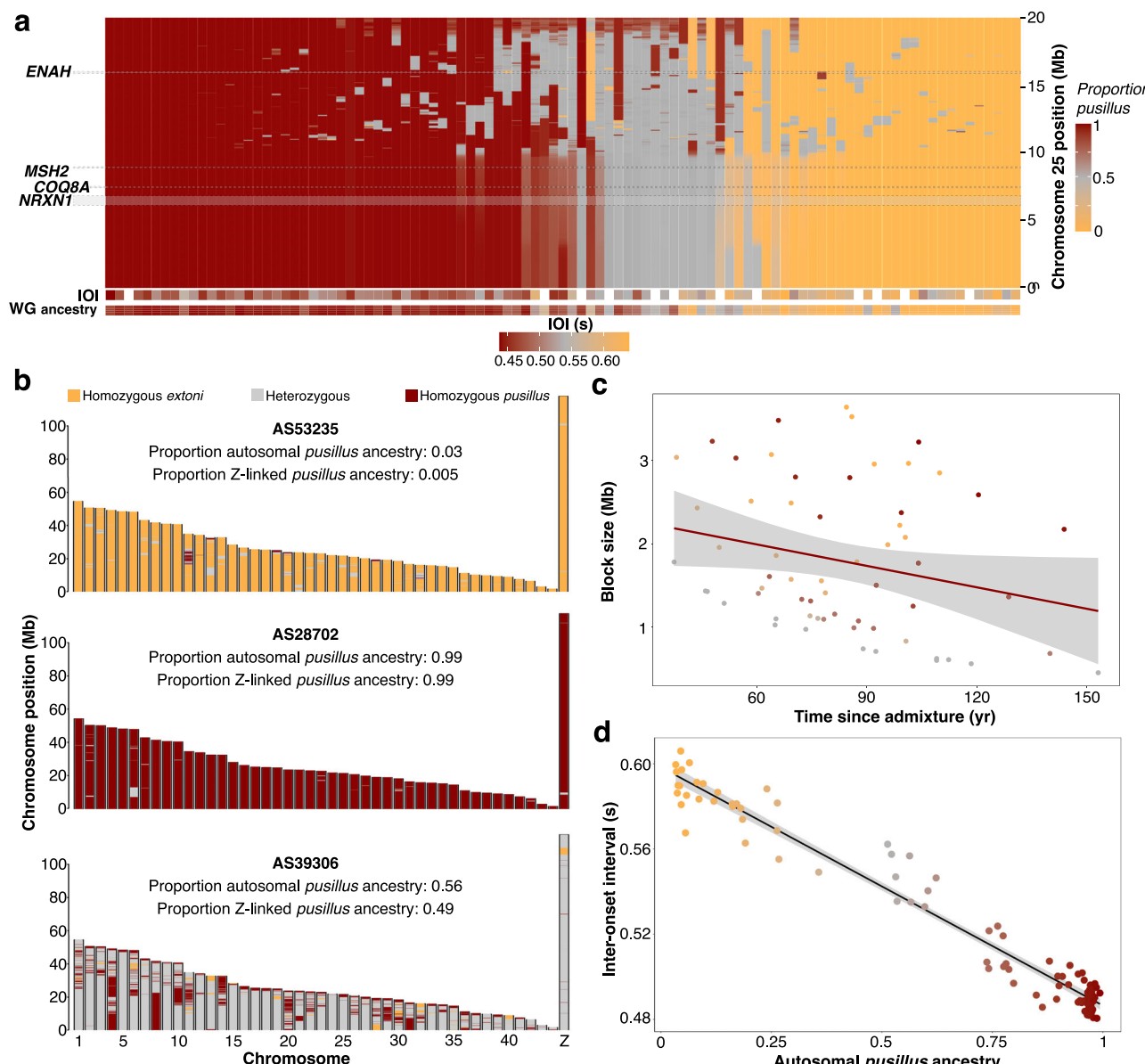

**Fig. 4 | The ancestry mosaicism of hybrid genomes.** Ancestry blocks illustrated **a** across chromosome 25 of hybrid zone individuals (vertical bars organized left to right from the most *pusillus* to the most *extoni* at chromosome 25) with respective IOI and whole-genome ancestry. Ancestry blocks across the entire genome of mostly one or other parental species and admixed ancestries are illustrated in (**b**), whereas the indirect relationship between block size and time since admixture is

represented in (**c**) (mean and 95% CI), with ancestry blocks becoming smaller the older the admixture event. The association between ancestry with IOI is represented in (**d**), where the black line represents the regression line of a linear mixed model (mean predicted values and 95% CI). Dot color in (**c**, **d**) follows the color-scheme used for *pusillus* ancestry in (**a**).

by linear mixed models (LMMs) in *glmmTMB*[52], which also confirmed no effect of sex and showed that higher proportions of autosomal and Z-linked *pusillus* ancestry were significantly associated with faster songs (Fig. 4d and Supplementary Table 10). Sigmoidal geographic clines for gene-specific ancestry also confirmed their association with IOI (Supplementary Fig. 11).

We next fitted DHGLMs to assess how ancestry proportions affect IOI, which we show is highly repeatable (Supplementary Methods 10−Estimation of repeatability) and its RWV. In addition to modeling a linear effect of ancestry on IOI stability, we also modeled quadratic effects on RWV to test the hypothesis that hybridization results in less

stable rhythmic song in admixed individuals. Consistent with the LMMs, greater *pusillus* ancestry was significantly associated with faster songs, but was also negatively associated with IOI RWV. This corresponds with more stable (i.e., less variance within) songs with increasing *pusillus* ancestry. However, the quadratic effect of ancestry on IOI RWV was not significant, but suggested higher variance (i.e., decreasing stability) with intermediate ancestry (Fig. 5a and Supplementary Table 11a−c). Yet, in the present study, we found that specific candidate genes affected song rhythm, and thus predicted that ancestry at those loci rather than genome-wide ancestry would more likely affect song stability. We tested for the effects of local ancestry

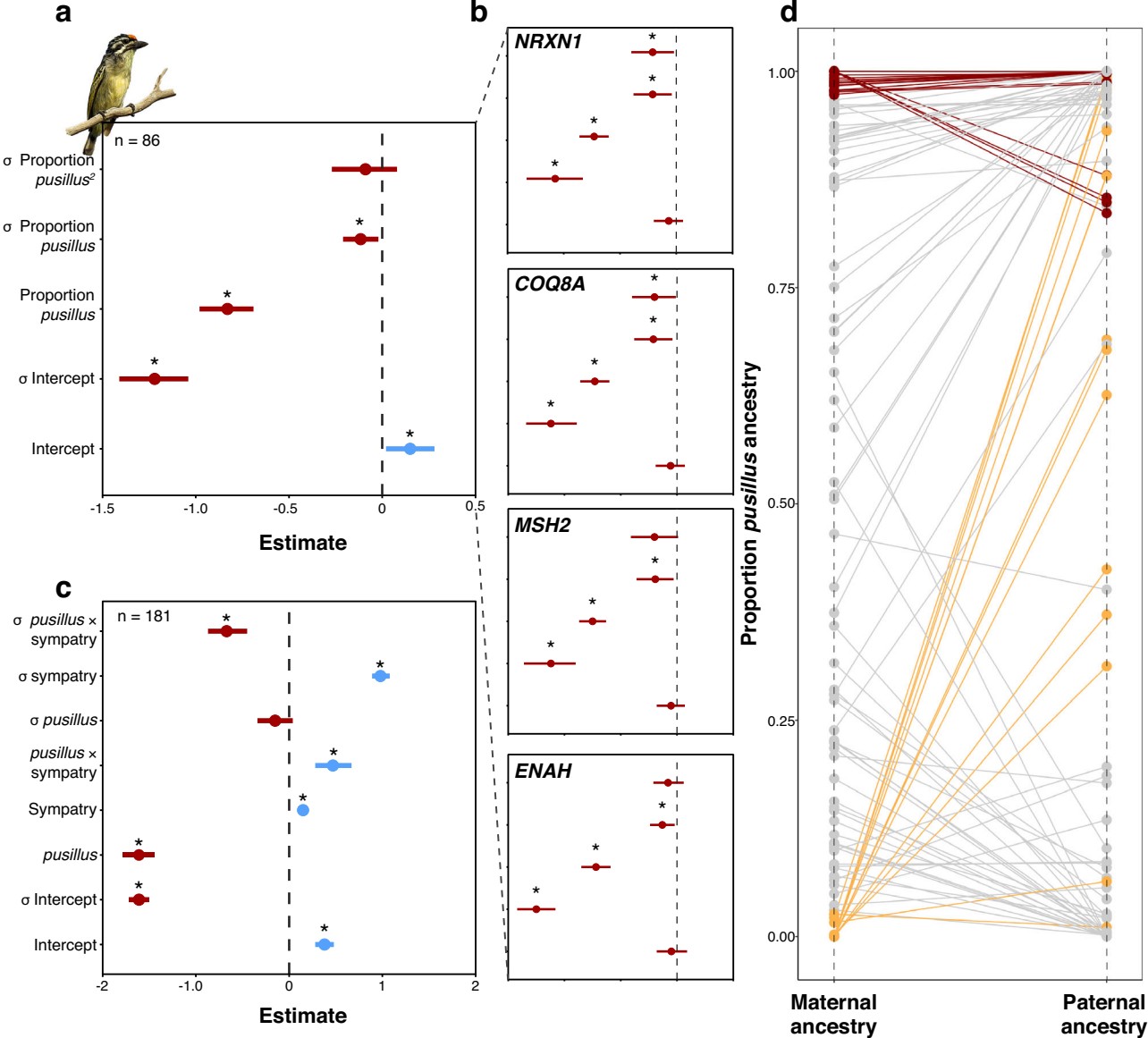

**Fig. 5 | Hybrid song instability, character displacement, and asymmetric introgression.** Double-hierarchical linear mixed model results illustrating **a** genome-wide ancestry effects on IOI and its variance (mean and 95% CI for 25,000 independent posterior draws), with higher *pusillus* ancestry associated with faster, and more stable songs, with the hypothesis of unstable songs in hybrids (H1) supported for those individuals with mixed ancestry (represented by quadratic term) specifically at **b** *NRXN1* and *COQ8A* (smaller panels). Support for character displacement **c** in song stability (H2) in pure parental species ancestry individuals (>99%), based on significant interaction of species (*pusillus*) and sympatry, showing that the difference in stability between the species is significantly greater in sympatry (represented by σ *pusillus* x sympatry) than in allopatry (represented by σ

*pusillus*), where it is not significantly different. σ terms indicate estimates for the variance part of the models, with * denoting statistically significant effects. Positive mean estimates and 95% CI (based on 37,500 independent posterior draws) are represented in blue, and negative in red. For exact estimates and 95% CI for the models in (**a**–**c**), please refer to Supplementary Tables 11, 13. Parental ancestries **d** of 95 females (heterogametic sex) in the contact zone, determined by assigning Z chromosome ancestry to fathers and calculating the proportion of the autosomal ancestry estimate attributed to mothers after accounting for paternal ancestry estimates. Pure *pusillus* mothers (>0.97 *pusillus* ancestry, red lines) mate assortatively with (<0.8) *pusillus* fathers, but pure *extoni* mothers (<0.03 *pusillus* ancestry, yellow lines) mate with males across the spectrum of *pusillus* ancestry.

proportions within the physical regions that contain the candidate genes associated with IOI. We averaged the ancestry proportion ($Q$) across all SNPs within the boundaries of the candidate genes on which significant SNPs mapped directly onto, and replicated the above model. We found that increasing proportions of *pusillus* ancestry at all four candidate genes were significantly associated with faster and more stable songs (Fig. 5b and Supplementary Table 11d–g). Moreover, in the case of *NRXN1* and *COQ8A* we found a significant quadratic effect on RWV that suggests individuals with admixture at these genes sing less stable songs. *Post hoc* analyses further suggested that individuals with higher *pusillus* ancestry at *NRXN1* and *COQ8A* are more sensitive to shifts in ancestry at these two genes. By contrast, individuals with primarily *extoni* ancestry, with increasing *pusillus* ancestry in *NRXN1* before the point of the maximum curvature of the quadratic effect (i.e., statistical peak) of RWV, have consistent IOI but with increasing variance. For *COQ8A*, the pre-peak effect was negative and marginally significant, suggesting increasing IOI with *extoni* ancestry but with a weaker effect from the statistical peak (Supplementary Fig. 12 and Supplementary Table 12).

### Possible character displacement in song stability

Intermediately paced or mixed rhythm songs might be unattractive to the opposite-sex[23] and, if hybrids then have lower fitness, drive reinforcement against hybridization[53]. We explored this possibility by testing for evidence of character displacement in IOI and its stability between 87 pure *extoni* and 94 *pusillus* individuals (respective ancestry >99%, based on fastSTRUCTURE ancestry values from double-digest restriction-site associated DNA (ddRAD) sequencing). Reproductive character displacement would be manifested in greater differences between the species in IOI or its variance in sympatry compared to allopatry. Instead of divergence in IOI, we found both species sang slower songs in sympatry. However, *extoni* emitted significantly less stable songs in sympatry than in allopatry, and we found a striking interaction effect of a greater difference in rhythmic stability between pure *pusillus* and *extoni* in sympatry compared with allopatry (Fig. 5c and Supplementary Table 13).

### Introgression is asymmetric in the tinkerbird hybrid zone

We investigated possible nonrandom mating by quantifying the direction of gene flow within the contact zone using 82,950 SNPs from ddRAD (mean depth = 13.92X, see Methods). We hypothesized that the presence of contrasting ancestry proportions between sex chromosomes and autosomes could reveal the direction of gene flow in hybridizing populations[54]. We bioinformatically inferred sex by calculating Z chromosome to autosome depth ratios[32] (Supplementary Fig. 13). Then, in females, the heterogametic sex in birds, we inferred paternal genome-wide ancestry from their paternally-inherited Z chromosome ancestry. Using the paternal ancestry estimate for 95 females, we calculated maternal ancestry from each female's autosomal ancestry (see Methods). There was clear evidence of asymmetric nonrandom mating (LM, $z = 6.949$, $P < 0.001$; Fig. 5d and Supplementary Table 14), consistent with previous findings revealing asymmetric introgression[32]. All 16 mothers with $Q_{pusillus} > 0.97$ mated with males with high *pusillus* proportions ($Q_{pusillus} > 0.83$, mean difference in parental ancestry $\Delta = 0.04 \pm 0.06$). By contrast, 14 mothers with $Q_{pusillus} < 0.03$ (i.e., *extoni* mothers) mated with males whose ancestry varied across the entire range of possible ancestry proportions ($\Delta = 0.62 \pm 0.34$), between mean ancestry proportions reported above genome-wide (0.6) and of 131 males sequenced with ddRAD from the hybrid zone ($Q_{pusillus} = 0.73 \pm 0.34$). These results suggest hybridization is asymmetric, with *extoni* females mating with males of any ancestry but *pusillus* females selecting males with high *pusillus* ancestry. Furthermore, although we observed a low correlation between IOI and body size ($r = 0.17$), and IOI and song frequency ($r = 0.28$) (Supplementary Methods 11—Assessing covariance of

phenotypes; Fig. 1b), there was a stronger correlation between IOI with forecrown hue ($r = 0.65$; Supplementary Fig. 14). Yet, red forecrowns (i.e., those with hue values <0.2) are not a peculiarity of individuals with higher proportions of *pusillus* ancestry, with mixed ancestry individuals typically sporting red forecrown feathers[32]. By contrast, faster songs (e.g., IOI values <0.5) are only sung by individuals with higher proportions of *pusillus* ancestry (Supplementary Fig. 14).

## Discussion

Our study has revealed a genomic basis for the speed of vocal rhythm. Pulse rate, represented here by IOI, is underpinned by at least four candidate genes with additive effects (Supplementary Fig. 8) located in a region of chromosome 25 of the tinkerbird genome. We further show that local introgression affects not only IOI, but its stability too, with individuals that have higher proportions of *pusillus* ancestry at these candidate genes singing both faster and rhythmically more consistent, and thus more stable songs. Moreover, differences in vocal consistency between pure *extoni* and *pusillus* in the hybrid zone are greater than they are in their respective allopatric populations, indicative of divergent character displacement in song stability. Divergent character displacement is a widely recognized mechanism for mediating reproductive isolation between related species[55,56]. Although, we do not have experimental evidence of the role rhythmic differences might play in mediating reproductive isolation among tinkerbirds, a pattern of asymmetric assortative mating in the contact zone suggests a preference in female *pusillus* for the *pusillus* phenotype in males, including their faster, more stable songs.

Many lines of research have attempted to unveil the genetic basis of bird song[13,15,16,57], but given that vocal learners are the predominant avian study system, providing candidate regions underlying innate vocal traits has been challenging. Thus far, *FOXP2* has been identified as the principal gene underlying song development through auditory feedback in vocal learning birds[58], and a recent study identified a set of 67 candidate genes associated with overall differences in song between two forms of presumed vocal non-learning suboscine *Empidonax* flycatchers, including some associated with song nuclei expressed in vocal learning and others with speech disorders in humans[57]. Our study goes beyond what is known so far by identifying candidate genes underlying one of the fundamental characteristics of bird vocalizations: their rhythmic pattern[19,59]. By focusing on a system that develops song innately, we identified *NRXN1*, *COQ8A*, *ENAH*, and *MSH2* as primary candidate genes underlying variation in IOI. Of these, *NRXN1* and *COQ8A* have both previously been associated with human speech disorders[60–62], and *ENAH* with hearing disabilities[63,64]. *MSH2* has yet to be associated with any vocal communication-related function, although it harbors the SNP most significantly associated with IOI, and it is mostly expressed in the cerebral cortex and nasopharynx in humans (www.proteinatlas.org)[65]. Among these candidate genes, we highlight *NRXN1*, with three significant SNPs mapping onto it, a gene that has been linked to several human neurological disorders, including autism[60]. Future work could focus on how a gene linked to impairments in speech and neurological disorders such as autism might also affect the speed and consistency of vocal rhythm.

Beyond genomic effects, the environment is thought to shape temporal, as well as spectral characters. After controlling for ancestry, we found no effect of vegetation density on pulse rate. This result was really not surprising, bearing in mind that allopatric *pusillus*, with its faster pulse rate, occupies much more densely vegetated habitats (i.e., coastal forest) than allopatric *extoni* that inhabits woodland savanna and sings slower-paced songs. The two species' respective pulse rates in allopatry are, therefore opposite to predictions of the acoustic adaptation hypothesis (AAH), suggesting that habitat might not constrain song pace, at least at the scales considered here. We did not consider the possible effects of beak size, which is known to affect the frequency and pulse rates of vocalizations[66,67]. Previous work did not

find a relationship between tinkerbird beak size and frequency[26]; besides, tinkerbirds do not open their beaks when they sing (Supplementary Movies 1–3); thus, beaks are unlikely to constrain their vocal performance.

We also inspected tinkerbird genomes to unveil the mosaicism of introgressed ancestry blocks, and showed that the continuous variation in IOI in tinkerbirds is influenced by ancestry at candidate genes, with such introgression affecting both IOI and its variance. Hybrids and individuals with more *extoni* ancestry sang slower-paced and more unstable songs than individuals with higher *pusillus* ancestry, with mixed ancestry at *NRXN1* and *COQ8A*, also affecting song stability. Greater rhythmic instability in hybrids could mirror the innate nature of the trait, whereby hybrid individuals sing intermediate and/or variably paced songs as a consequence of their admixed genomes, a result that is in agreement with a proposed genetic basis for song variability[68]. Also, instability in hybrid song may affect mate choice as well if females prefer males with more stable songs. A preference for fast, stable song in *pusillus* invokes the concept of female choice based on male motor performance[69], an indicator of male quality[70] that has been shown to drive mate choice in passerines[23], and might therefore underpin asymmetric assortative mating in the tinkerbird hybrid zone. This hypothesis is supported by a pattern of divergent character displacement in rhythmic stability between *pusillus* and *extoni* in sympatry compared to allopatry.

The pattern of asymmetric hybridization we show here intertwines perfectly with the above findings: the higher instability in pure *extoni* and hybrid individuals could result in a preference for the faster and more stable songs produced by pure *pusillus*. However, many other factors could contribute to this mating asymmetry. Body size, for instance, which correlates with song frequency[25], has been shown to drive assortative mating in Darwin's finches[37] and the covariance of either of the two with IOI could obscure inference. Nevertheless, although we show there is no apparent correlation between IOI and either song frequency or body mass, there is a relationship between IOI and tinkerbird forecrown hue, for which a previous study associated a region on chromosome 8 with carotenoid conversion[32]. Indeed, besides in song, *extoni* and *pusillus* mostly differ from human observers in the coloration of their forecrown[32], as their common names suggest, and such visual communication signals have been widely shown to influence mating preferences and thus could be an important pre-mating barrier to gene flow[71].

Strikingly, although in the hybrid zone all individuals with higher *pusillus* ancestry sing both faster songs and display red forecrown feathers, the two traits are decoupled in admixed birds, which typically sport red forecrown feathers[32] but sing slower songs. Based on the pattern of nonrandom mating we present here, *pusillus* females did not mate with such mixed ancestry individuals with red forecrowns. This suggests that fast, stable songs may be the critical component for female *pusillus*, or it could be the combination of red forecrown plumage with fast, stable songs. In any case, *pusillus* females do not appear to discriminate between males on the basis of forecrown color alone. We caveat that these interpretations are based on the correspondence of parental ancestries we calculated and the distribution of song and forecrown color traits in the hybrid zone. Further support for the role of rhythmic characters in mate choice would require experimental study. Besides phenotypic traits, the relative densities of the two species and other factors, including possible genetic incompatibilities such as cytonuclear incompatibilities[72], may all contribute to the apparent nonrandom mating pattern we found in this southern African contact zone. Further work would thus be needed to investigate the relative contribution of these phenotypic traits as well as other intrinsic or extrinsic factors as possible barriers to gene flow.

We also revealed isochronous vocalizations in birds whose songs are innate, with tinkerbirds delivering notes at equally spaced time intervals. Isochrony, which has been identified in humans, non-human primates, songbirds, and bats[24,30,73], is thought to facilitate acoustic coordination and processing, especially in vocal learners[24]. Tinkerbirds are not vocal learners[74], but contrary to assertions that vocal non-learners lack the ability to perceive isochronous rhythms[75], our study suggests its perception[74] might have driven selection for rhythmic song components even in species with limited vocal flexibility. Indeed, although tinkerbirds do not duet, perception of rhythmic patterning is likely fundamental for duet coordination in confamilial species from independent clades within Lybiidae that perform highly coordinated multi-individual acoustic displays[36].

Our study thus takes us a step forward in our understanding of the relationship between the molecular basis of bird song, the genomics of avian vocal rhythm, isochrony, and the development of musicality in humans[19,76]. Our results complement recent findings that have advanced our understanding of the genetic basis of musicality in humans, bird vocalizations[29,57,77], and the links between them[78], including the identification of genetic regions associated with differences in beat synchronization and, therefore, rhythm perception in humans[29]. Moreover, areas in the brain underlying such beat synchronization also underlie vocal learning in birds, thus supporting a framework of a common genetic basis for rhythm between humans and songbirds[78]. We propose candidate genes underlying rhythm speed in Piciform birds that develop innate songs: *NRXN1*, *COQ8A*, *ENAH*, and *MSH2* (but also genes with SNPs mapping in their proximity: e.g. *ALK*, *FOXN2*, and *EML4*, Supplementary Table 5) and show that phenotypic differences may also arise because of introgression of ancestry blocks in specific genomic regions. There is the possibility that the same or similar regions could function in rhythm in both vocal non-learners and learners; indeed, regions similar to the pallial song nuclei of vocal learners increase in expression in the forebrains of downy woodpecker (*Picoides pubescens*) during their drumming display[79], a rhythmic signal used for mate attraction and territory defense[80]. We eagerly anticipate future studies into the genetics underlying rhythm-related behaviors across vocal non-learners and learners that may unveil the mechanisms of a potentially shared musical heritage with humans.

## Methods

### Ethics statement

In accordance with the Nagoya Protocol on Access to Genetic Resources and the Fair and Equitable Sharing of Benefits Arising from their Utilization to the Convention on Biological Diversity, this study was conducted in accordance with all ethical guidelines: SAFRING permit 15966; Kingdom of Swaziland/Eswatini Big Game Parks No. PI 0673, PI 0982, PI 1022, PI 1117, PI 1160, PI 1161, and PI 1205, Ezemvelo KwaZulu-Natal PRN: OP 4645/2015 and PRN: HO/4068/02, Mpumalanga Tourism and Parks Agency permits MPB. 5526 and MPB. 5626, and Gauteng permit CPF6-000217 and the Animal Research Ethics Committee of the University of KwaZulu-Natal (AREC/00002381/2021 and AREC/00002382/2021). Furthermore, researchers from countries where genetic resources were obtained were collaborators in the study and included as co-authors, aliquots of genetic material collected during this study were provided to local institutions, and knowledge generated by this study was shared with local stakeholders.

### Fieldwork

Fieldwork was performed in Eswatini and South Africa between 2015 and 2022 to sample and record the allopatric and sympatric populations of yellow-fronted tinkerbird *Pogoniulus chrysoconus extoni* and red-fronted tinkerbird *Pogoniulus pusillus pusillus*. Our sampling efforts focused on the breeding season - the rainy season—to take advantage of the territorial response of breeding pairs. Tinkerbirds were lured into mist nets using conspecific playbacks, measured, and provided with a uniquely numbered metal band and a specific combination of color bands prior to release. As part of our sampling

protocol, we also plucked forecrown feathers for spectral reflectance analysis (see ref. 32 for spectral reflectance details and Supplementary Methods 1—Sample collection and storage). We visited capture sites repeatedly thereafter, with the aim of locating color-banded tinkerbirds, which would then be elicited to sing with the use of conspecific playbacks, which have a negligible effect on IOI: (see Supplementary Methods 9 and Supplementary Fig. 15 for a comparison of IOI before and after playback). We recorded vocal responses of the color-banded birds, although we also recorded unbanded tinkerbirds as well every time we were presented with the opportunity, aiming for ~3 min of recordings (~120 notes per minute in *P. pusillus*) using a Marantz PMD 661 with either a Sennheiser MKH 8050 directional super-cardioid microphone or MKH 8070 long shotgun microphone and saved recordings as 16-bit WAV files at a sampling frequency of 48 kHz. In total, we sampled 468 tinkerbirds and obtained 710 recordings from 491 individuals (mean $2.2 \pm 1.5$ sd recordings per individual) across allopatric and sympatric sites. A subset of these recordings was previously used for another study focusing on continent-wide patterns of song frequency[26].

### Acoustic analyses

WAV files were imported into Raven Pro v1.6[81], where notes (mean $388 \pm$ sd 380 notes per individual) were detected using the built-in band-limited energy detectors (BLED) following a previously established protocol[26,82]. We calculated the difference between the onset times of two consecutive notes to obtain the inter-onset interval (IOI). This workflow resulted in two datasets, one containing all IOI intervals (hereafter "All Notes") and one with IOI values averaged across each individual recording (hereafter "mean IOI").

### Quantification of rhythmicity in tinkerbirds

To assess rhythmicity in tinkerbird song we quantified categorical rhythms, in which temporal intervals among notes are distributed discreetly rather than continuously[30,83]. Such rhythm categories can be identified by calculating rhythmic ratios ($R_k$)[24], in which each IOI value is divided by itself plus the value of the subsequent interval ($R_k = IOI_k/(IOI_K + IOI_{K+1})$), where lowercase "k" indicates a specific note interval. Following this approach, an $R_k$ of 0.5 would equate to a 1:1 rhythm ratio and, therefore, isochrony, meaning that each note is delivered at an equally spaced interval.

### Genomic data extraction, library preparation, and sequencing

We generated a de novo chromosome-level reference genome assembly from a *pusillus* female individual sampled in Mlawula Game Reserve, Eswatini, using four different sequencing technologies: Pacific Biosciences (PacBio) CLR long-reads, 10x Genomics linked-reads, Bionano optical maps and Hi-C reads from Arima Genomics (see Supplementary Methods 2 for a detailed description).

### Chromosome-level reference genome assembly and evaluation

Prior to the assembly, a *k*-mer histogram (31 bp) was generated with Meryl v1.4.1[41] from unassembled 10x linked-reads generated for the reference genome individual. The histogram was then used with Genomescope[84] to estimate genome size, heterozygosity, and repeat content. The reference genome was assembled with the VGP standard genome assembly pipeline 1.6[40] using the PacBio CLR long reads, 10x linked-reads, Bionano optical maps, and Hi-C reads (see Supplementary Methods 3 for a detailed description).

### Whole-genome variant calling and double-digest restriction-site associated DNA sequencing (ddRAD)

We processed raw whole-genome sequences of 137 *extoni* and *pusillus* (Supplementary Data File 1) from allopatric and sympatric populations in GATK (see Supplementary Methods 4 for a detailed description of the variant calling pipeline). To investigate the direction of

backcrossing, we complemented the whole-genome dataset with ddRADs since most individuals sequenced at the whole-genome level were males. SNPs were called from 452 samples of *extoni* and *pusillus* (Supplementary Data File 2) following a standardized pipeline using Stacks[85] v. 2.62 (Supplementary Methods 5).

### Relative sequence depth sexing

Male birds have two copies of the Z chromosome, while females have one Z and one W chromosome. We therefore compared the relative sequence depth of the Z chromosome in each individual to autosome depth, with the expectation that males would have a similar Z chromosome to autosome depth with two copies of each, whereas females would have half the depth on the Z compared to autosomes[32]. Using this approach, female individuals should have a depth ratio centered around 0.5, whereas males should have a value of approximately 1. Based on the depth ratio distribution, we classified individuals with a *Z:autosome* depth ratio <0.7 as females and those with a *Z:autosome* depth ratio >0.9 as males.

### Recombination rate estimates

To aid our interpretation of the genomic landscape of differentiation, we estimated recombination rates ($\rho$) using the *interval* command in LDhat 2.2[86] using a reference panel of 26 allopatric *extoni*. LDhat calculates the likelihood surface for different recombination rates for each pair of SNPs by simulating coalescent trees under a reversible jump Markov chain Monte Carlo (rjMCMC) scheme. The likelihoods between SNPs are combined with the likelihood of a region using a composite likelihood method. The recombination rates with the highest likelihoods are then chosen for each region. We allowed the rjMCMC to run for five million iterations using a block penalty of 10 and sampling every 5000 iterations.

### Population structure analyses

To investigate population structure across our samples, we first created an unlinked dataset by pruning our MAF-filtered whole-genome dataset to filter for loci within 100 Kb windows distance with an $r^2$ above 0.1 (using *--indep-pairwise 100 kb 10 0.1*) in PLINK v1.90b3i[87]. Principal Components Analysis (PCA) was then conducted on allele frequencies across the entire genome in PLINK using the *--pca* flag. Furthermore, we performed PCA on specific genomic regions identified by the analysis in GEMMA (see *Identifying candidate loci*) to investigate population structuring within the loci that are significantly associated with IOI. For analysis of genome-wide population structure we used ADMIXTURE v1.3[88] and, for consistency with other studies on this Southern African population[31], set the assumed number of populations (i.e., K) equal to 2.

We also inferred population structure from our ddRAD dataset by running fastSTRUCTURE[89]. This was done to obtain a measure of ancestry from more individuals within the population and, specifically, more females, since only a portion of the total individuals sampled were sequenced at the whole-genome level with priority given to color-banded individuals whose songs we recorded, which were mostly males. To determine the direction of introgression in the hybrid zone, which previous studies suggested was asymmetric[32], we needed to focus on the heterogametic sex, i.e. females, because paternal ancestry could be estimated based on the ancestry of the single Z chromosome inherited from the father. To do so, we compared ancestry values calculated in fastSTRUCTURE between the Z chromosome and the autosomes (see *Assessing the direction of backcrossing* for a detailed description).

### Identifying candidate loci underlying IOI

To identify candidate regions that underpin differences in IOI between *extoni* and *pusillus* we ran linear mixed-effects models (LMMs, using the *-lmm 1* command) and BSLMMs in GEMMA v0.98[46]. We focused

specifically on 87 color-banded individuals recorded and sequenced at the whole-genome level from the contact zone only, to minimize the effects of spatial population structuring (i.e., isolation-by-distance) and accounted for potential relatedness among individuals by supplying an estimated relatedness matrix as a covariate in the LMM and BSLMM (for further details see Supplementary Methods 6−Identifying candidate loci underlying IOI).

### Efficient local ancestry inference

We further investigated local ancestry for each individual in the contact zone using a two-layer hidden Markov model implemented in the efficient local ancestry inference (ELAI) software[90]. This approach utilizes linkage disequilibrium within and between parental populations and assigns dosage scores between 0 and 2 (for a two-way admixture model) that reflect ancestry proportions in each SNP in individuals from admixed populations. Dosage scores of 0 and 2 indicate each homozygous state, whereas a dosage score of 1 reveals the heterozygous state.

After identifying parental populations based on sampling localities (allopatric sites) and ADMIXTURE scores (see Population structure analysis), we applied two upper-layer clusters (-C) and ten lower-layer clusters (-c) (five times the value of -C, as recommended in the user's manual). Given the uncertainty related to the timing of the admixture event, we investigated four possible values of the admixture generations parameter (i.e., -mg), therefore estimating ancestry scores assuming five, ten, fifteen, and twenty generations since the admixture event. For each migration parameter, we ran three independent runs with 30 EM steps (-s) and then averaged the 12 independent runs. We classified sites with allele dosage scores between 0.5 and 1.5 as heterozygous, sites <0.5 as homozygous for *extoni*, and those with scores >1.5 as homozygous for *pusillus* alleles. We further used ELAI output to estimate admixture times (i.e., the first allele-sharing event between parental species) in our hybrid individuals (see Supplementary Methods 7−Estimating timing of admixture in the contact zone).

### Assessing the direction of backcross

To estimate possible asymmetry of introgression in the hybrid zone between *extoni* and *pusillus*, we compared the extent of admixture in the autosomes and the Z chromosome separately in 95 females sequenced with ddRADseq using fastSTRUCTURE. By inheriting only one copy of the Z chromosome from their fathers, female birds can be used to determine the direction of backcrossing in hybrids. This can be achieved by comparing the Z chromosome and the autosomal ancestry. Assuming that the Z chromosome ancestry of a female individual reflects her father's Z chromosome ancestry, that her autosomal ancestry is an average of her two parents' ancestry value, and that the father's autosomal ancestry equals his Z chromosome ancestry, then the maternal ancestry = (2 x autosomal ancestry) - Z ancestry. Using a similar approach to ref. 54, we quantified differences in mating preferences by fitting a linear model (LM) using the difference between the calculated ancestries for each parent (Δ parental ancestry) as a response variable and a binary variable for maternal ancestry, pure (>97%) *pusillus* ($n = 16$) vs. pure (>97%) *extoni* ($n = 14$) as the predictor, expecting pure *pusillus* mothers to have a significantly lower Δ parental ancestry as a consequence of the more similar ancestry between parents. Since female preferences for forecrown plumage color, body size, or frequency may also affect mating patterns in the hybrid zone, we evaluate whether IOI covaries with these relevant phenotypes by estimating Pearson's correlations (Supplementary Methods 11−Assessing covariance of phenotypes).

### Genome scans for diversity and linkage

Using the no-MAF dataset with invariant sites, we calculated genome-wide $F_{ST}$, a relative measure of genetic differentiation, $D_{XY}$, an absolute measure of genetic differentiation, and $\pi$, nucleotide diversity,

between allopatric *extoni* and *pusillus* in 25 kb non-overlapping windows using the *popgenWindows.py* custom script (https://github.com/simonhmartin/genomics_general). To shed light on whether concordance between relatively high $F_{ST}$ and $D_{XY}$ peaks arises as a consequence of gene flow or whether it results from ancient divergence of haplotypes, we also ran the above genomic scans with sympatric individuals that exceeded 95% pure ancestry. This selection resulted in an unbalanced sample size, consisting of 30 sympatric individuals with *pusillus* ancestry >95%, but only eight individuals with *extoni* ancestry >95%. We therefore performed these scans by randomly extracting eight *pusillus* individuals >95% and then running the genome scan. We performed this process 100 times and then averaged the $p$ values from the 100 scans (see Supplementary Fig. 5b, c).

For graphical purposes, we also computed genome-wide $F_{ST}$ in 500 kb windows (see Fig. 2) as well as $r^2$, a measure of linkage disequilibrium (LD). To estimate the latter, we first calculated LD in 25 kb non-overlapping windows and, following ref. 91, we averaged genome-wide LD values within 500 kb blocks using the *chr_ld.pl* custom script (https://github.com/SwallowGenomics/BarnSwallow/blob/main/Analyses/LD-scripts/chr_ld.pl) to visualize areas of the genome that are in high linkage (see Fig. 2).

### Genome scans for signatures of selection

Genome-wide diversity statistics such as $F_{ST}$ and $D_{XY}$ can be prone to bias, and are specifically affected by variation in recombination rates across the genome. Hence, we also calculated cross-population extended haplotype homozygosity (xpEHH) in allopatric individuals, a method that compares haplotype lengths between populations to control for local variation in recombination rates[49]. Increasing frequencies of selected alleles result in long haplotypes surrounding the selected allele in a population that has undergone a selective sweep. By comparing the haplotype homozygosity between two lineages, xpEHH can identify regions that have undergone selective sweeps and, therefore subject to positive selection. This method therefore allowed us to identify genomic signatures of positive selection at those loci underpinning IOI and, by consequence, selection for faster or slower songs. In our case, highly positive values indicate selection in *extoni*, whereas negative values indicate selection in *pusillus*. To calculate xpEHH, we first phased each chromosome in the MAF-filtered dataset separately using SHAPEIT v4.2[92] and then computed xpEHH across allopatric individuals using the *rehh* v3.2.2 package in R[93].

### Geographic clines

We fitted sigmoidal geographic clines to assess how variation in IOI and ancestry relates to distance from the contact zone. To do so, we used ArcMap v10.7 to calculate the distance of each data point to a line drawn through the center of the contact zone. Since such a line extends from East to West, we attributed negative distance values to points south of the contact line and positive distance values to points north of this line. To avoid large sampling gaps on either side of the contact line, for this analysis, we also included recordings of unbanded tinkerbirds from allopatric sites in each parental species' distribution, which we assumed have pure *extoni* or *pusillus* ancestry respectively (proportion *pusillus* = 0 for allopatric *extoni* and proportion *pusillus* = 1 for allopatric *pusillus*). We then fitted geographic clines for IOI and ancestry in the *HZAR* R package v0.2[94] following the scripts provided therein.

### Statistical analyses−testing for hybrid song instability

We further investigated the relationship between ancestry and IOI by running linear mixed-effects models in the *glmmTMB* R package v1.1.8[52]. We set IOI as a dependent variable and the proportion of *pusillus* ancestry and sex as fixed effects, since the seven individuals we recorded were females. We ran separate models for autosomal and

Z-linked proportions of *pusillus* ancestry, with individual ID nested in sampling location as random effects. Model fit was validated using the functions provided in the *DHARMa* v.0.4.6 package in R[95].

We also investigated the relationship between ancestry and IOI from a variance perspective to determine how ancestry may affect the stability of rhythmic song. To achieve this, we used univariate double-hierarchical generalized linear models (DHGLMs) to estimate random and fixed effects in both the mean and residual within-individual variance (RWV) parts of models[96] using *brms* v.2.20.2 in R[97]. We used the 'All Notes' dataset to investigate the impact of hybridization on tinkerbird IOI, with a particular focus on the effects of mixed ancestry on IOI RWV.

We hypothesized that because tinkerbirds develop their songs innately[74], hybrids might be expected to share acoustic features of both parental species as a consequence of their admixed genomes, thus resulting in more temporally unstable songs. Pure ancestry individuals would thus sing less variable, more temporally consistent songs than hybrids. To test for the effects of ancestry on IOI and its stability, we modeled IOI as a function of the proportion of *pusillus* ancestry across the entire genome for 86 individuals with individual ID as a random factor. We used the same predictors and random effect in the RWV part of the model plus a quadratic function for *pusillus* ancestry. We did this to model a possible non-linear relationship between variance in IOI and ancestry, with intermediate values of ancestry predicted to have higher RWV than extremes representing pure ancestry of either species if hybrids emitted less stable songs. Moreover, in addition to using whole-genome ancestry values, we aimed to pinpoint possible effects in candidate genes by using ancestry proportions from the specific genes associated with IOI in GEMMA that were extracted with BLAST and their ancestry calculated in ELAI. We therefore replicated the best-selected model with whole-genomic ancestry four times, each time replacing whole-genome ancestry with gene-specific ancestry (i.e., one independent model for each gene). For any model with a significant quadratic effect (i.e., with 95% CI not overlapping zero), we also ran an additional post hoc analysis in *lme4* v.1.1[98] to investigate the patterns of the quadratic effect (see Supplementary Methods 8—Hybrid song instability models and post hoc test).

### Statistical analyses—testing for character displacement in IOI

We also hypothesized that if divergent character displacement through reinforcement has a stabilizing effect on IOI, then pure ancestry individuals in the contact zone are expected to have more stable songs than individuals in allopatry. We added acoustic data from recordings of 59 unbanded *extoni* and 38 *pusillus* individuals from distant allopatry and assumed *pusillus* ancestry of 0 and 1, respectively, for *extoni* and *pusillus* based on estimated Q values from fastSTRUCTURE v1.0 of 83 *extoni* ($0.01 \pm 0.1$) and 24 *pusillus* ($0.99 \pm 0.003$) from allopatric populations. We fitted a model on the resulting dataset of 181 individuals with ancestry estimates $Q < 0.01$ for *extoni* ($n = 87$) and $Q > 0.99$ for *pusillus* ($n = 94$), incorporating 84 individuals with ancestry values estimated in fastSTRUCTURE. We used standardized IOI as the response variable in DHGLMs, and included species, population (categorical with two levels: allopatric vs. sympatric), and their interaction, as fixed factors, with individual ID used as a random factor. This structure was used in both the "mean" and the "RWV" parts of the model. For these models, we ran five chains over 7500 iterations (750 warm-up), and the maximal tree depth was set to 15. Model convergence and chain mixing for all the above-mentioned models were evaluated using the *Rhat* estimates and by graphical inspection of the trace plots.

### Statistical analyses—habitat structure and IOI

The temporal patterning of acoustic signals may be influenced by habitat type in accordance with the acoustic adaptation hypothesis (AAH)[35]. The AAH posits that slower-paced songs are favored in more densely vegetated habitats, given that the higher reverberation of sounds in such environments could result in suboptimal communication. To account for this possibility, we extracted environmental remote-sensing data for all our recording localities. Specifically, we extracted the enhanced vegetation index product (EVI), a measure of canopy structure and vegetation greenness from the MODIS Terra satellite (see ref. 26 for details on remote-sensing data extraction). This allowed us to test for an effect of EVI on IOI, and therefore account for any effect of vegetation according to the AAH. We fitted generalized linear mixed models in *glmmTMB* using IOI as the response variable, while using EVI and whole-genome proportion of *pusillus* ancestry as fixed factors and location as a random factor to account for potential habitat differences across sympatric sites. Models were validated with *DHARMa* using the functions provided therein.

## Data availability

All sequencing data associated with this study have been deposited in the GenBank database under accession codes GCA_015220805.1 and GCA_015220175.1, BioProject accession number PRJNA987636. Processed genomic data are available on Figshare https://figshare.com/articles/dataset/A_genomic_basis_of_vocal_rhythm_in_birds/25308376[99]. Source data are provided as a Source Data file. Source data are provided with this paper.

## Code availability

Scripts used for the analyses are available on GitHub https://github.com/MatteoSebastianelli/Tinkerbird_SongGene[100].

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

## Acknowledgements

We thank EC Nwankwo, P Simelane, S Khoza, M Mamba, and KG Mortega for fieldwork, E Fourie, A Howland, M and S McGinn, J and R Harding, S Agostini, G Vogt, J Visser, L Klapwijk, R Olivier, and M Greve for logistical support, and BO Ogolowa and M Garrigos for bioinformatics. Eswatini Big Game Parks, Ezemvelo KZN Wildlife, Mpumalanga Tourism and Parks Agency, Gauteng Department of Agriculture and Rural Development and the Animal Research Ethics Committee of the University of KwaZulu-Natal (reference number: AREC/00002381-82/2021), the Cyprus Ministry of Agriculture, rural development and environment veterinary ser-

vices, and USDA provided research permits. This study was possible thanks to funding from the European Regional Development Fund and the Republic of Cyprus through the Research and Innovation Foundation (Project: EXCELLENCE/0421/0301, A.N.G.K.), University of Cyprus Research Grant No. 8037P-25023 (A.N.G.K.), FP7 Marie Curie Reintegration Grant No. 268316 (A.N.G.K.), AG Leventis Foundation grants (M.S., S.M.L., M.M., and S.G.d.S.) and American Ornithological Society covid relief fund (M.S.).

## Author contributions

Authors contributed to the manuscript as follows: Conceptualization: M.S., A.B., B.M.v.H., and A.N.G.K. Methodology: M.S., S.M.L., S.S., A.H., N.J.D., E.D.J., A.B., B.M.v.H., and A.N.G.K. Genome assembly and curation: B.H., J.M., J.B., S.P., W.C., O.F., and E.D.J. Fieldwork planning and logistics: C.T.D., A.M., and A.N.G.K. Fieldwork: S.M.L., M.S., A.M., A.B., and A.N.G.K. Molecular labwork: M.M. and C.N. Investigation: M.S., S.M.L., S.S., S.G.d.S., A.B., B.M.v.H., and A.N.G.K. Visualization: M.S., S.S. Funding acquisition: A.N.G.K., M.S., and S.M.L. Project administration: A.N.G.K. Supervision: A.N.G.K., A.B., and B.M.v.H. Writing—original draft: M.S., S.S., B.M.v.H., A.B., and A.N.G.K. Writing—review and editing: All authors.

## Funding

## Competing interests

The authors declare no competing interests.
