## [Peer Review File · Nature Communications]

A genomic basis of vocal rhythm in birdsThis manuscript has been previously reviewed at another journal that is not operating a transparent peer review scheme. This document only contains reviewer comments and rebuttal letters for versions considered at *Nature Communications*.

REVIEWERS' COMMENTS

Reviewer #1 (Remarks to the Author):

This is an exciting and comprehensive review of the potential genes underlying a phenotype related to vocal rhythm. Here, the authors provide information that an aspect of avian song – its rhythm – is a key trait mediating hybrid matings when two species of tinkerbird have overlapping distributions. A key objective of the paper (as stated) is to determine the genetic basis of this trait. I reviewed a previous draft of this manuscript and find the revision to be greatly improved.

In particular, the conceptual framework driving the work and a directed hypothesis are more clearly stated. I found it much easier to understand how the trait in question - vocal rhythm - was measured and quantified.

The authors do an excellent job of highlighting the primary contribution of this paper - the genetic associations with a behavioral trait that appears to mediate reproductive isolation.

I appreciate the work that went into the revision process!

****Reviewer 1's assessment of authors responses to Reviewer 3 (upon request from Editor):**

First, a summary of reviewer 3's comments and my sense for whether they were addressed in the revision. My answer is: partially. I think there could be greater transparency around the focus on chromosome 25.

Reviewer 3 highlighted several concerns related to the genetic analyses and whether SNP investigations were as transparent and thorough as possible.

While the authors have emphasized the whole genome approach to analyzing the genetic landscape of divergence, it is in fact the case that they focus on a subset of the locations detected in the genome-wide analyses, with a primary focus on chromosome 25. I think some additional language can be added here to be more direct about this focus. Based on Fig 3, there does seem to be a lot of activity on Chromosome 25, but there are others that have interesting looking regions as well: chromosomes 3, 5, and 8. Was it the case that there are no interesting loci in there after inspection or was the investigation solely focused on chromosome 25 and if so, why? There is no wrong answer here but some information about the approach should be stated otherwise it does feel like the authors are highlighting only the coolest results instead of sharing a result that could look like: of four chromosomes with elevated regions of differentiation, we found candidate loci associated with the trait of interest on chromosome 25 alone.

The reviewer brought up the issue related to false discovery rates, and I believe this was satisfactorily addressed.

Finally, reviewer 3 brought up comments related to the inferences about mate selection and the overall claims being made. I had the same responses to the first draft and I feel that they have been satisfactorily addressed. The authors are clearer about the exploratory nature of their study and are more tempered in their approach to assigning a process-based understanding of genomic differentiation (e.g., that selection on genes on chromosome 25 are associated with sexual selection and are barriers to gene flow).

Reviewer #2 (Remarks to the Author):

All the comments and suggestions I made after the first time I reviewed this manuscript were properly addressed by the authors, thanks.

After reading this new version I only have two very minor comments:

L178-180: Authors first say they extracted 76 SNPs with the largest sparse effect on IOI, and in the following sentences they refer to "these 67 SNPs". Is this a typo? Are they referring to a subgroup of 67 out of those 76 SNPs?

Some references are all in lowercase, such as 10., and 39. There may be more, please check.

RESPONSE TO REVIEWERS' COMMENTS

Reviewer #1 (Remarks to the Author):

This is an exciting and comprehensive review of the potential genes underlying a phenotype related to vocal rhythm. Here, the authors provide information that an aspect of avian song – its rhythm – is a key trait mediating hybrid matings when two species of tinkerbird have overlapping distributions. A key objective of the paper (as stated) is to determine the genetic basis of this trait. I reviewed a previous draft of this manuscript and find the revision to be greatly improved.

In particular, the conceptual framework driving the work and a directed hypothesis are more clearly stated. I found it much easier to understand how the trait in question - vocal rhythm - was measured and quantified.

The authors do an excellent job of highlighting the primary contribution of this paper - the genetic associations with a behavioral trait that appears to mediate reproductive isolation.

I appreciate the work that went into the revision process!

****Reviewer 1's assessment of authors responses to Reviewer 3 (upon request from Editor):**

First, a summary of reviewer 3's comments and my sense for whether they were addressed in the revision. My answer is: partially. I think there could be greater transparency around the focus on chromosome 25.

Reviewer 3 highlighted several concerns related to the genetic analyses and whether SNP investigations were as transparent and thorough as possible.

While the authors have emphasized the whole genome approach to analyzing the genetic landscape of divergence, it is in fact the case that they focus on a subset of the locations detected in the genome-wide analyses, with a primary focus on chromosome 25. I think some additional language can be added here to be more direct about this focus. Based on Fig 3, there does seem to be a lot of activity on Chromosome 25, but there are others that have interesting looking regions as well: chromosomes 3, 5, and 8. Was it the case that there are no interesting loci in there after inspection or was the investigation solely focused on chromosome 25 and if so, why? There is no wrong answer here but some information about the approach should be stated otherwise it does feel like the authors are highlighting only the coolest results instead of sharing a result that could look like: of four chromosomes with elevated regions of differentiation, we found candidate loci associated with the trait of interest on chromosome 25 alone. We have now made clear in the main text (L154-155) that all the analyses have focused on genomic areas that had more than one SNP associated with the inter-onset interval.

The reviewer brought up the issue related to false discovery rates, and I believe this was satisfactorily addressed.

Finally, reviewer 3 brought up comments related to the inferences about mate selection and the overall claims being made. I had the same responses to the first draft and I feel that they have been satisfactorily addressed. The authors are clearer about the exploratory nature of their study and are more tempered in their approach to assigning a process-based understanding of genomic differentiation (e.g., that selection on genes on chromosome 25 are associated with sexual selection and are barriers to gene flow).

Reviewer #2 (Remarks to the Author):

All the comments and suggestions I made after the first time I reviewed this manuscript were properly addressed by the authors, thanks.

After reading this new version I only have two very minor comments:

L178-180: Authors first say they extracted 76 SNPs with the largest sparse effect on IOI, and in the following sentences they refer to "these 67 SNPs". Is this a typo? Are they referring to a subgroup of 67 out of those 76 SNPs?

We thank reviewer two for noticing this inconsistency. We realized this was simply a typo, with the corrected value being 76. We edited the manuscript accordingly (L180-181)

Some references are all in lowercase, such as 10., and 39. There may be more, please check.

We have now fixed reference 10 and 39 and ensured that all the other references are properly cited.